# Spectral Learning of Dynamic Systems from Nonequilibrium Data

**Hao Wu and Frank Noé**
Department of Mathematics and Computer Science
Freie Universität Berlin
Arnimallee 6, 14195 Berlin
{hao.wu,frank.noe}@fu-berlin.de

## Abstract

Observable operator models (OOMs) and related models are one of the most important and powerful tools for modeling and analyzing stochastic systems. They exactly describe dynamics of finite-rank systems and can be efficiently and consistently estimated through spectral learning under the assumption of identically distributed data. In this paper, we investigate the properties of spectral learning without this assumption due to the requirements of analyzing large-time scale systems, and show that the equilibrium dynamics of a system can be extracted from nonequilibrium observation data by imposing an equilibrium constraint. In addition, we propose a binless extension of spectral learning for continuous data. In comparison with the other continuous-valued spectral algorithms, the binless algorithm can achieve consistent estimation of equilibrium dynamics with only linear complexity.

## 1 Introduction

In the last two decades, a collection of highly related dynamic models including observable operator models (OOMs) [1–3], predictive state representations [4–6] and reduced-rank hidden Markov models [7, 8], have become powerful and increasingly popular tools for analysis of dynamic data. These models are largely similar, and all can be learned by spectral methods in a general framework of multiplicity automata, or equivalently sequential systems [9, 10]. In contrast with the other commonly used models such as Markov state models [11, 12], Langevin models [13, 14], traditional hidden Markov models (HMMs) [15, 16], Gaussian process state-space models [17, 18] and recurrent neural networks [19], the spectral learning based models can exactly characterize the dynamics of a stochastic system without any a priori knowledge except the assumption of finite dynamic rank (i.e., the rank of Hankel matrix) [10, 20], and the parameter estimation can be efficiently performed for discrete-valued systems without solving any intractable inverse or optimization problem. We focus in this paper only on stochastic systems without control inputs and all spectral learning based models can be expressed in the form of OOMs for such systems, so we will refer to them as OOMs below.

In most literature on spectral learning, the observation data are assumed to be identically (possibly not independently) distributed so that the expected values of observables associated with the parameter estimation can be reliably computed by empirical averaging. However, this assumption can be severely violated due to the limit of experimental technique or computational capacity in many practical situations, especially where metastable physical or chemical processes are involved. A notable example is the distributed computing project Folding@home [21], which explores protein folding processes that occur on the timescales of microseconds to milliseconds based on molecular dynamics simulations on the order of nanoseconds in length. In such a nonequilibrium case where distributions of observation data are time-varying and dependent on initial conditions, it is still unclear

if promising estimates of OOMs can be obtained. In [22], a hybrid estimation algorithm was proposed to improve spectral learning of large-time scale processes by using both dynamic and static data, but it still requires assumption of identically distributed data. One solution to reduce the statistical bias caused by nonequilibrium data is to discard the observation data generated before the system reaches steady state, which is a common trick in applied statistics [23]. Obviously, this way suffers from substantial information loss and is infeasible when observation trajectories are shorter than mixing times. Another possible way would be to learn OOMs by likelihood-based estimation instead of spectral methods, but there is no effective maximum likelihood or Bayesian estimator of OOMs until now. The maximum pseudo-likelihood estimator of OOMs proposed in [24] demands high computational cost and its consistency is yet unverified.

Another difficulty for spectral approaches is learning with continuous data, where density estimation problems are involved. The density estimation can be performed by parametric methods such as the fuzzy interpolation [25] and the kernel density estimation [8]. But these methods would reduce the flexibility of OOMs for dynamic modeling because of their limited expressive capacity. Recently, a kernel embedding based spectral algorithm was proposed to cope with continuous data [26], which avoids explicit density estimation and learns OOMs in a nonparametric manner. However, the kernel embedding usually yields a very large computational complexity, which greatly limits practical applications of this algorithm to real-world systems.

The purpose of this paper is to address the challenge of spectral learning of OOMs from nonequilibrium data for analysis of both discrete- and continuous-valued systems. We first provide a modified spectral method for discrete-valued stochastic systems which allows us to consistently estimate the equilibrium dynamics from nonequilibrium data, and then extend this method to continuous observations in a binless manner. In comparison with the existing learning methods for continuous OOMs, the proposed binless spectral method does not rely on any density estimator, and can achieve consistent estimation with linear computational complexity in data size even if the assumption of identically distributed observations does not hold. Moreover, some numerical experiments are provided to demonstrate the capability of the proposed methods.

## 2 Preliminaries

### 2.1 Notation

In this paper, we use $\mathbb{P}$ to denote probability distribution for discrete random variables and probability density for continuous random variables. The indicator function of event $e$ is denoted by $1_e$ and the Dirac delta function centered at $x$ is denoted by $\delta_x(\cdot)$. For a given process $\{a_t\}$, we write the subsequence $(a_k, a_{k+1}, \ldots, a_{k'})$ as $a_{k:k'}$, and $\mathbb{E}_\infty[a_t] \triangleq \lim_{t \to \infty} \mathbb{E}[a_t]$ means the equilibrium expected value of $a_t$ if the limit exists. In addition, the convergence in probability is denoted by $\xrightarrow{p}$.

### 2.2 Observable operator models

An $m$-dimensional observable operator model (OOM) with observation space $\mathcal{O}$ can be represented by a tuple $\mathcal{M} = (\boldsymbol{\omega}, \{\boldsymbol{\Xi}(x)\}_{x \in \mathcal{O}}, \boldsymbol{\sigma})$, which consists of an initial state vector $\boldsymbol{\omega} \in \mathbb{R}^{1 \times m}$, an evaluation vector $\boldsymbol{\sigma} \in \mathbb{R}^{m \times 1}$ and an observable operator matrix $\boldsymbol{\Xi}(x) \in \mathbb{R}^{m \times m}$ associated to each element $x \in \mathcal{O}$. $\mathcal{M}$ defines a stochastic process $\{x_t\}$ in $\mathcal{O}$ as

$$\mathbb{P}(x_{1:t}|\mathcal{M}) = \boldsymbol{\omega}\boldsymbol{\Xi}(x_{1:t})\boldsymbol{\sigma} \tag{1}$$

under the condition that $\boldsymbol{\omega}\boldsymbol{\Xi}(x_{1:t})\boldsymbol{\sigma} \geq 0$, $\boldsymbol{\omega}\boldsymbol{\Xi}(\mathcal{O})\boldsymbol{\sigma} = 1$ and $\boldsymbol{\omega}\boldsymbol{\Xi}(x_{1:t})\boldsymbol{\sigma} = \boldsymbol{\omega}\boldsymbol{\Xi}(x_{1:t})\boldsymbol{\Xi}(\mathcal{O})\boldsymbol{\sigma}$ hold for all $t$ and $x_{1:t} \in \mathcal{O}^t$ [10], where $\boldsymbol{\Xi}(x_{1:t}) \triangleq \boldsymbol{\Xi}(x_1)\ldots\boldsymbol{\Xi}(x_t)$ and $\boldsymbol{\Xi}(\mathcal{A}) \triangleq \int_{\mathcal{A}} \mathrm{d}x\, \boldsymbol{\Xi}(x)$. Two OOMs $\mathcal{M}$ and $\mathcal{M}'$ are said to be equivalent if $\mathbb{P}(x_{1:t}|\mathcal{M}) \equiv \mathbb{P}(x_{1:t}|\mathcal{M}')$.

## 3 Spectral learning of OOMs

### 3.1 Algorithm

Here and hereafter, we only consider the case that the observation space $\mathcal{O}$ is a finite set. (Learning with continuous observations will be discussed in Section 4.2.) A large number of largely similar

---

**Algorithm 1** General procedure for spectral learning of OOMs

---

**INPUT:** Observation trajectories generated by a stochastic process $\{x_t\}$ in $\mathcal{O}$

**OUTPUT:** $\hat{\mathcal{M}} = (\hat{\omega}, \{\hat{\Xi}(x)\}_{x \in \mathcal{O}}, \hat{\sigma})$

**PARAMETER:** $m$: dimension of the OOM. $D_1, D_2$: numbers of feature functions. $L$: order of feature functions.

1: Construct feature functions $\phi_1 = (\varphi_{1,1}, \ldots, \varphi_{1,D_1})^\top$ and $\phi_2 = (\varphi_{2,1}, \ldots, \varphi_{2,D_2})^\top$, where each $\varphi_{i,j}$ is a mapping from $\mathcal{O}^L$ to $\mathbb{R}$ and $D_1, D_2 \geq m$.

2: Approximate

$$\bar{\phi}_1 \triangleq \mathbb{E}\left[\phi_1(x_{t-L:t-1})\right], \quad \bar{\phi}_2 \triangleq \mathbb{E}\left[\phi_2(x_{t:t+L-1})\right] \tag{5}$$

$$\mathbf{C}_{1,2} \triangleq \mathbb{E}\left[\phi_1(x_{t-L:t-1})\phi_2(x_{t:t+L-1})^\top\right] \tag{6}$$

$$\mathbf{C}_{1,3}(x) \triangleq \mathbb{E}\left[1_{x_t=x} \cdot \phi_1(x_{t-L:t-1})\phi_2(x_{t+1:t+L})^\top\right], \quad \forall x \in \mathcal{O} \tag{7}$$

by their empirical means $\hat{\bar{\phi}}_1$, $\hat{\bar{\phi}}_2$, $\hat{\mathbf{C}}_{1,2}$ and $\hat{\mathbf{C}}_{1,3}(x)$ over observation data.

3: Compute $\mathbf{F}_1 = \mathbf{U}\boldsymbol{\Sigma}^{-1} \in \mathbb{R}^{D_1 \times m}$ and $\mathbf{F}_2 = \mathbf{V} \in \mathbb{R}^{D_2 \times m}$ from the truncated singular value decomposition $\hat{\mathbf{C}}_{1,2} \approx \mathbf{U}\boldsymbol{\Sigma}\mathbf{V}^\top$, where $\boldsymbol{\Sigma} \in \mathbb{R}^{m \times m}$ is a diagonal matrix contains the top $m$ singular values of $\hat{\mathbf{C}}_{1,2}$, and $\mathbf{U}$ and $\mathbf{V}$ consist of the corresponding $m$ left and right singular vectors of $\hat{\mathbf{C}}_{1,2}$.

4: Compute

$$\hat{\sigma} = \mathbf{F}_1^\top \hat{\bar{\phi}}_1 \tag{8}$$

$$\hat{\Xi}(x) = \mathbf{F}_1^\top \hat{\mathbf{C}}_{1,3}(x)\mathbf{F}_2, \quad \forall x \in \mathcal{O} \tag{9}$$

$$\hat{\omega} = \hat{\bar{\phi}}_2^\top \mathbf{F}_2 \tag{10}$$

---

spectral methods have been developed, and the generic learning procedure of these methods is summarized in Algorithm 1 by omitting details of algorithm implementation and parameter choice [27, 7, 28]. For convenience of description and analysis, we specify in this paper the formula for calculating $\hat{\bar{\phi}}_1$, $\hat{\bar{\phi}}_2$, $\hat{\mathbf{C}}_{1,2}$ and $\hat{\mathbf{C}}_{1,3}(x)$ in Line 2 of Algorithm 1 as follows:

$$\hat{\bar{\phi}}_1 = \frac{1}{N}\sum_{n=1}^{N}\phi_1(\vec{s}_n^1), \quad \hat{\bar{\phi}}_2 = \frac{1}{N}\sum_{n=1}^{N}\phi_2(\vec{s}_n^2) \tag{2}$$

$$\hat{\mathbf{C}}_{1,2} = \frac{1}{N}\sum_{n=1}^{N}\phi_1(\vec{s}_n^1)\phi_2(\vec{s}_n^2)^\top \tag{3}$$

$$\hat{\mathbf{C}}_{1,3}(x) = \frac{1}{N}\sum_{n=1}^{N}1_{s_n^2=x}\phi_1(\vec{s}_n^1)\phi_2(\vec{s}_n^3)^\top, \quad \forall x \in \mathcal{O} \tag{4}$$

Here $\{(\vec{s}_n^1, s_n^2, \vec{s}_n^3)\}_{n=1}^{N}$ is the collection of all subsequences of length $(2L+1)$ appearing in observation data ($N = T - 2L$ for a single observation trajectory of length $T$). If an observation subsequence $x_{t-L:t+L}$ is denoted by $(\vec{s}_n^1, s_n^2, \vec{s}_n^3)$ with some $n$, then $\vec{s}_n^1 = x_{t-L:t-1}$ and $\vec{s}_n^3 = x_{t+1:t+L}$ represents the prefix and suffix of $x_{t-L:t+L}$ of length $L$, $s_n^2 = x_t$ is the intermediate observation value, and $\vec{s}_n^2 = x_{t:t+L-1}$ is an "intermediate part" of the subsequence of length $L$ starting from time $t$ (see Fig. 1 for a graphical illustration).

Algorithm 1 is much more efficient than the commonly used likelihood-based learning algorithms and does not suffer from local optima issues. In addition, and more importantly, this algorithm can be shown to be consistent if $(\vec{s}_n^1, s_n^2, \vec{s}_n^3)$ are (i) independently sampled from $\mathcal{M}$ or (ii) obtained from a finite number of trajectories which have fully mixed so that all observation triples are identically distributed (see, e.g., [8, 3, 10] for related works). However, the asymptotic correctness of OOMs learned from short trajectories starting from nonequilibrium states has not been formally determined.

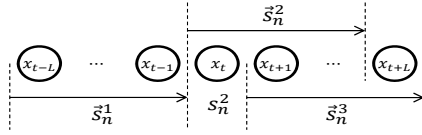

Figure 1: Illustration of variables $\vec{s}_n^1$, $s_n^2$, $\vec{s}_n^3$ and $\vec{s}_n^2$ used in Eqs. (2)-(4) with $(\vec{s}_n^1, s_n^2, \vec{s}_n^3) = x_{t-L:t+L}$.

## 3.2 Theoretical analysis

We now analyze statistical properties of the spectral algorithm without the assumption of identically distributed observations. Before stating our main result, some assumptions on observation data are listed as follows:

**Assumption 1.** *The observation data consists of $I$ independent trajectories of length $T$ produced by a stochastic process $\{x_t\}$, and the data size tends to infinity with (i) $I \to \infty$ and $T = T_0$ or (ii) $T \to \infty$ and $I = I_0$.*

**Assumption 2.** *$\{x_t\}$ is driven by an $m$-dimensional OOM $\mathcal{M} = (\boldsymbol{\omega}, \{\boldsymbol{\Xi}(x)\}_{x \in \mathcal{O}}, \boldsymbol{\sigma})$, and*

$$\frac{1}{T'} \sum_{t=1}^{T'} f_t \xrightarrow{p} \mathbb{E}_\infty \left[ f\left(x_{t:t+l-1}\right) \right] = \mathbb{E}_\infty \left[ f\left(x_{t:t+l-1}\right) | x_{1:k} \right] \tag{11}$$

*as $T' \to \infty$ for all $k$, $l$, $x_{1:k}$ and $f : \mathcal{O}^l \mapsto \mathbb{R}$.*

**Assumption 3.** *The rank of the limit of $\hat{\mathbf{C}}_{1,2}$ is not less than $m$.*

Notice that Assumption 2 only states the asymptotic stationarity of $\{x_t\}$ and marginal distributions of observation triples are possibly time dependent if $\boldsymbol{\omega} \neq \boldsymbol{\omega}\boldsymbol{\Xi}(\mathcal{O})$. Assumption 3 ensures that the limit of $\hat{\mathcal{M}}$ given by Algorithm 1 is well defined, which generally holds for minimal OOMs (see [10]).

Based on the above assumptions, we have the following theorem concerning the statistical consistency of the OOM learning algorithm (see Appendix A.1 for proof):

**Theorem 1.** *Under Assumptions 1-3, there exists an OOM $\mathcal{M}' = (\boldsymbol{\omega}', \{\boldsymbol{\Xi}'(x)\}_{x \in \mathcal{O}}, \boldsymbol{\sigma}')$ which is equivalent to $\hat{\mathcal{M}}$ and satisfies*

$$\boldsymbol{\sigma}' \xrightarrow{p} \boldsymbol{\sigma}, \quad \boldsymbol{\Xi}'(x) \xrightarrow{p} \boldsymbol{\Xi}(x), \; \forall x \in \mathcal{O} \tag{12}$$

This theorem is central in this paper, which implies that the spectral learning algorithm can achieve consistent estimation of all parameters of OOMs except initial state vectors even for nonequilibrium data. ($\hat{\boldsymbol{\omega}} \xrightarrow{p} \boldsymbol{\omega}'$ does not hold in most cases except when $\{x_t\}$ is stationary.). It can be further generalized according to requirements in more complicated situations where, for example, observation trajectories are generated with multiple different initial conditions (see Appendix A.2).

## 4 Spectral learning of equilibrium OOMs

In this section, applications of spectral learning to the problem of recovering equilibrium properties of dynamic systems from nonequilibrium data will be highlighted, which is an important problem in practice especially for thermodynamic and kinetic analysis in computational physics and chemistry.

### 4.1 Learning from discrete data

According to the definition of OOMs, the equilibrium dynamics of an OOM $\mathcal{M} = (\boldsymbol{\omega}, \{\boldsymbol{\Xi}(x)\}_{x \in \mathcal{O}}, \boldsymbol{\sigma})$ can be described by an equilibrium OOM $\mathcal{M}_{\text{eq}} = (\boldsymbol{\omega}_{\text{eq}}, \{\boldsymbol{\Xi}(x)\}_{x \in \mathcal{O}}, \boldsymbol{\sigma})$ as

$$\lim_{t \to \infty} \mathbb{P}\left(x_{t+1:t+k} = z_{1:k} | \mathcal{M}\right) = \mathbb{P}\left(x_{1:t} = z_{1:k} | \mathcal{M}_{\text{eq}}\right) \tag{13}$$

if the equilibrium state vector

$$\boldsymbol{\omega}_{\text{eq}} = \lim_{t \to \infty} \boldsymbol{\omega}\boldsymbol{\Xi}(\mathcal{O})^t \tag{14}$$

exists. From (13) and (14), we have

$$\begin{cases} \boldsymbol{\omega}_{\text{eq}}\boldsymbol{\Xi}(\mathcal{O}) = \lim_{t\to\infty} \boldsymbol{\omega}_{\text{eq}}\boldsymbol{\Xi}(\mathcal{O})^{t+1} = \boldsymbol{\omega}_{\text{eq}} \\ \boldsymbol{\omega}_{\text{eq}}\boldsymbol{\sigma} = \lim_{t\to\infty} \sum_{x\in\mathcal{O}} \mathbb{P}\left(x_{t+1} = x\right) = 1 \end{cases} \tag{15}$$

The above equilibrium constraint of OOMs motivates the following algorithm for learning equilibrium OOMs: *Perform Algorithm 1 to get $\hat{\boldsymbol{\Xi}}(x)$ and $\hat{\boldsymbol{\sigma}}$ and calculate $\hat{\boldsymbol{\omega}}_{\text{eq}}$ by a quadratic programming problem*

$$\hat{\boldsymbol{\omega}}_{\text{eq}} = \arg\min_{\mathbf{w}\in\{\mathbf{w}|\mathbf{w}\hat{\boldsymbol{\sigma}}=1\}} \left\| \mathbf{w}\hat{\boldsymbol{\Xi}}(\mathcal{O}) - \mathbf{w} \right\|^2 \tag{16}$$

(See Appendix A.3 for a closed-form expression of the solution to (16).)

The existence and uniqueness of $\boldsymbol{\omega}_{\text{eq}}$ are shown in Appendix A.3, which yield the following theorem:

**Theorem 2.** *Under Assumptions 1-3, the estimated equilibrium OOM $\hat{\mathcal{M}}_{\text{eq}} = (\hat{\boldsymbol{\omega}}_{\text{eq}}, \{\hat{\boldsymbol{\Xi}}(x)\}_{x\in\mathcal{O}}, \hat{\boldsymbol{\sigma}})$ provided by Algorithm 1 and Eq. (16) satisfies*

$$\mathbb{P}\left(x_{1:l} = z_{1:l}|\hat{\mathcal{M}}_{\text{eq}}\right) \xrightarrow{p} \lim_{t\to\infty} \mathbb{P}\left(x_{t+1:t+l} = z_{1:l}\right) \tag{17}$$

*for all $l$ and $z_{1:l}$.*

*Remark* 1. $\hat{\boldsymbol{\omega}}_{\text{eq}}$ can also be computed as an eigenvector of $\hat{\boldsymbol{\Xi}}(\mathcal{O})$. But the eigenvalue problem possibly yields numerical instability and complex values because of statistical noise, unless some specific feature functions $\boldsymbol{\phi}_1, \boldsymbol{\phi}_2$ are selected so that $\hat{\boldsymbol{\omega}}_{\text{eq}}\hat{\boldsymbol{\Xi}}(\mathcal{O}) = \hat{\boldsymbol{\omega}}_{\text{eq}}$ can be exactly solved in the real field [29].

## 4.2   Learning from continuous data

A straightforward way to extend spectral algorithms to handle continuous data is based on the coarse-graining of the observation space. Suppose that $\{x_t\}$ is a stochastic process in a continuous observation space $\mathcal{O} \subset \mathbb{R}^d$, and $\mathcal{O}$ is partitioned into $J$ discrete bins $\mathcal{B}_1, \ldots, \mathcal{B}_J$. Then we can utilize the algorithm in Section 4.1 to approximate the equilibrium transition dynamics between bins as

$$\lim_{t\to\infty} \mathbb{P}\left(x_{t+1} \in \mathcal{B}_{j_1}, \ldots, x_{t+l} \in \mathcal{B}_{j_l}\right) \approx \hat{\boldsymbol{\omega}}_{\text{eq}}\hat{\boldsymbol{\Xi}}(\mathcal{B}_{j_1}) \ldots \hat{\boldsymbol{\Xi}}(\mathcal{B}_{j_l})\hat{\boldsymbol{\sigma}} \tag{18}$$

and obtain a *binned* OOM $\hat{\mathcal{M}}_{\text{eq}} = (\hat{\boldsymbol{\omega}}_{\text{eq}}, \{\hat{\boldsymbol{\Xi}}(x)\}_{x\in\mathcal{O}}, \hat{\boldsymbol{\sigma}})$ for the continuous dynamics of $\{x_t\}$ with

$$\hat{\boldsymbol{\Xi}}(x) = \frac{\hat{\boldsymbol{\Xi}}(\mathcal{B}(x))}{\text{vol}(\mathcal{B}(x))} \tag{19}$$

by assuming the observable operator matrices are piecewise constant on bins, where $\mathcal{B}(x)$ denotes the bin containing $x$ and $\text{vol}(\mathcal{B})$ is the volume of $\mathcal{B}$. Conventional wisdom dictates that the number of bins is a key parameter for the coarse-graining strategy and should be carefully chosen for the balance of statistical noise and discretization error. However, we will show in what follows that it is justifiable to increase the number of bins to infinity.

Let us consider the limit case where $J \to \infty$ and bins are infinitesimal with $\max_j \text{vol}(\mathcal{B}_j) \to 0$. In this case,

$$\hat{\boldsymbol{\Xi}}(x) = \lim_{\text{vol}(\mathcal{B}(x))\to 0} \frac{\hat{\boldsymbol{\Xi}}(\mathcal{B}(x))}{\text{vol}(\mathcal{B}(x))} = \begin{cases} \hat{\mathbf{W}}_{s_n^2}\delta_{s_n^2}(x), & x = s_n^2 \\ 0, & \text{otherwise} \end{cases} \tag{20}$$

where

$$\hat{\mathbf{W}}_{s_n^2} = \frac{1}{N}\mathbf{F}_1^\top \boldsymbol{\phi}_1(\vec{s}_n^{\,1})\boldsymbol{\phi}_2(\vec{s}_n^{\,3})^\top \mathbf{F}_2 \tag{21}$$

according to (9) in Algorithm 1. Then $\hat{\mathcal{M}}_{\text{eq}}$ becomes a *binless* OOM over sample points $\mathcal{X} = \{s_n^2\}_{n=1}^N$ and can be estimated from data by Algorithm 2, where the feature functions can be selected as indicator functions, radial basis functions or other commonly used activation functions for single-layer neural networks in order to digest adequate dynamic information from observation data.

The binless algorithm presented here can be efficiently implemented in a linear computational complexity $O(N)$, and is applicable to more general cases where observations are strings, graphs or other structured variables. Unlike the other spectral algorithms for continuous data, it does not require

**Algorithm 2** Procedure for learning binless equilibrium OOMs

---

**INPUT:** Observation trajectories generated by a stochastic process $\{x_t\}$ in $\mathcal{O} \subset \mathbb{R}^d$

**OUTPUT:** Binless OOM $\hat{\mathcal{M}} = (\hat{\boldsymbol{\omega}}, \{\hat{\boldsymbol{\Xi}}(x)\}_{x \in \mathcal{O}}, \hat{\boldsymbol{\sigma}})$

1: Construct feature functions $\boldsymbol{\phi}_1 : \mathbb{R}^{Ld} \mapsto \mathbb{R}^{D_1}$ and $\boldsymbol{\phi}_2 : \mathbb{R}^{Ld} \mapsto \mathbb{R}^{D_2}$ with $D_1, D_2 \geq m$.

2: Calculate $\hat{\bar{\boldsymbol{\phi}}}_1, \hat{\bar{\boldsymbol{\phi}}}_2, \hat{\mathbf{C}}_{1,2}$ by (2) and (3).

3: Compute $\mathbf{F}_1 = \mathbf{U}\boldsymbol{\Sigma}^{-1} \in \mathbb{R}^{D_1 \times m}$ and $\mathbf{F}_2 = \mathbf{V} \in \mathbb{R}^{D_2 \times m}$ from the truncated singular value decomposition $\hat{\mathbf{C}}_{1,2} \approx \mathbf{U}\boldsymbol{\Sigma}\mathbf{V}^\top$.

4: Compute $\hat{\boldsymbol{\sigma}}, \hat{\boldsymbol{\omega}}$ and $\hat{\boldsymbol{\Xi}}(x) = \sum_{z \in \mathcal{X}} \hat{\mathbf{W}}_z \delta_z(x)$ by (8), (16) and (21), where $\hat{\boldsymbol{\Xi}}(\mathcal{O}) = \int_\mathcal{O} \mathrm{d}x \, \hat{\boldsymbol{\Xi}}(x) = \sum_{z \in \mathcal{X}} \hat{\mathbf{W}}_z$.

---

that the observed dynamics coincides with some parametric model defined by feature functions. Lastly but most importantly, as stated in the following theorem, this algorithm can be used to consistently extract static and kinetic properties of a dynamic system in equilibrium from nonequilibrium data (see Appendix A.3 for proof):

**Theorem 3.** *Provided that the observation space $\mathcal{O}$ is a closed set in $\mathbb{R}^d$, feature functions $\boldsymbol{\phi}_1, \boldsymbol{\phi}_2$ are bounded on $\mathcal{O}^L$, and Assumptions 1-3 hold, the binless OOM given by Algorithm 2 satisfies*

$$\mathbb{E}\left[g\left(x_{1:r}\right) | \hat{\mathcal{M}}_{\mathrm{eq}}\right] \xrightarrow{p} \mathbb{E}_\infty\left[g\left(x_{t+1:t+r}\right)\right] \tag{22}$$

*with*

$$\mathbb{E}\left[g\left(x_{1:r}\right) | \hat{\mathcal{M}}_{\mathrm{eq}}\right] = \sum_{x_{1:r} \in \mathcal{X}^r} g\left(x_{1:r}\right) \hat{\boldsymbol{\omega}} \hat{\mathbf{W}}_{z_1} \ldots \hat{\mathbf{W}}_{z_r} \hat{\boldsymbol{\sigma}} \tag{23}$$

   *(i) for all continuous functions $g : \mathcal{O}^r \mapsto \mathbb{R}$.*

   *(ii) for all bounded and Borel measurable functions $g : \mathcal{O}^r \mapsto \mathbb{R}$, if there exist positive constants $\bar{\xi}$ and $\underline{\xi}$ so that $\|\boldsymbol{\Xi}(x)\| \leq \bar{\xi}$ and $\lim_{t \to \infty} \mathbb{P}\left(x_{t+1:t+r} = z_{1:r}\right) \geq \underline{\xi}$ for all $x \in \mathcal{O}$ and $z_{1:r} \in \mathcal{O}^r$.*

### 4.3 Comparison with related methods

It is worth pointing out that the spectral learning investigated in this section is an ideal tool for analysis of dynamic properties of stochastic processes, because the related quantities, such as stationary distributions, principle components and time-lagged correlations, can be easily computed from parameters of discrete OOMs or binless OOMs. For many popular nonlinear dynamic models, including Gaussian process state-space models [17] and recurrent neural networks [19], the computation of such quantities is intractable or time-consuming.

The major disadvantage of spectral learning is that the estimated OOMs are usually only "approximately valid" and possibly assign "negative probabilities" to some observation sequences. So it is difficult to apply spectral methods to prediction, filtering and smoothing of signals where the Bayesian inference is involved.

## 5 Applications

In this section, we evaluate our algorithms on two diffusion processes and the molecular dynamics of alanine dipeptide, and compare them to several alternatives. The detailed settings of simulations and algorithms are provided in Appendix B.

**Brownian dynamics** Let us consider a one-dimensional diffusion process driven by the Brownian dynamics

$$\mathrm{d}x_t = -\nabla V(x_t)\mathrm{d}t + \sqrt{2\beta^{-1}}\mathrm{d}W_t \tag{24}$$

with observations generated by

$$y_t = \begin{cases} 1, & x_t \in \mathrm{I} \\ 0, & x_t \in \mathrm{II} \end{cases}$$

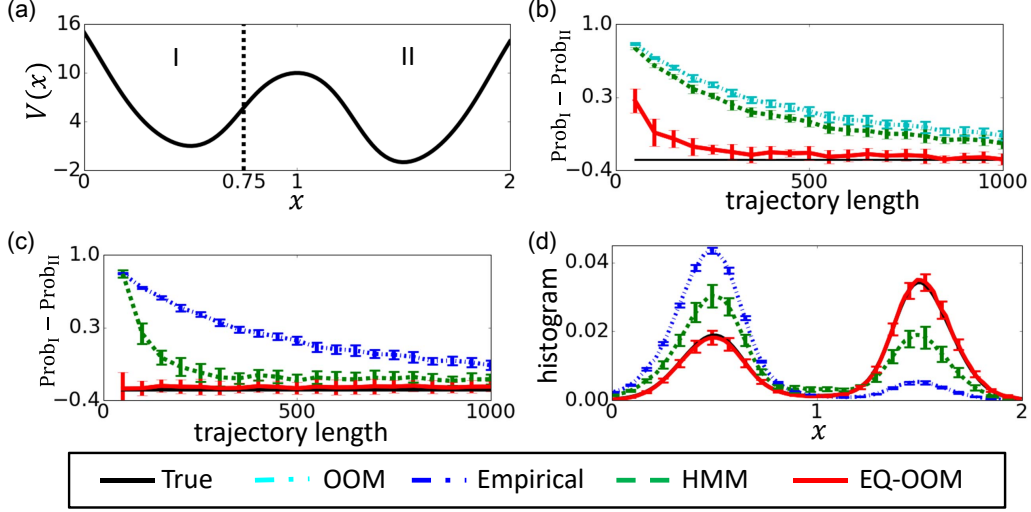

Figure 2: Comparison of modeling methods for a one-dimensional diffusion process. (a) Potential function. (b) Estimates of the difference between equilibrium probabilities of I and II given by the traditional OOM, HMM and the equilibrium OOM (EQ-OOM) obtained from the proposed algorithm with $\mathcal{O} = \{\mathrm{I}, \mathrm{II}\}$. (c) Estimates of the probability difference given by the empirical estimator, HMM and the proposed binless OOM with $\mathcal{O} = [0, 2]$. (d) Stationary histograms of $\{x_t\}$ with 100 uniform bins estimated from trajectories with length 50. The length of each trajectory is $T = 50 \sim 1000$ and the number of trajectories is $[10^5/T]$. Error bars are standard deviations over 30 independent experiments.

The potential function $V(x)$ is shown in Fig. 2(a), which contains two potential wells I, II. In this example, all simulations are performed by starting from a uniform distribution on $[0, 0.2]$, which implies that simulations are highly nonequilibrium and it is difficult to accurately estimate the equilibrium probabilities $\mathrm{Prob}_{\mathrm{I}} = \mathbb{E}_\infty [1_{x_t \in \mathrm{I}}] = \mathbb{E}_\infty [y_t]$ and $\mathrm{Prob}_{\mathrm{II}} = \mathbb{E}_\infty [1_{x_t \in \mathrm{II}}] = 1 - \mathbb{E}_\infty [y_t]$ of the two potential wells from the simulation data. We first utilize the traditional spectral learning without enforcing equilibrium, expectation–maximization based HMM learning and the proposed discrete spectral algorithm to estimate $\mathrm{Prob}_{\mathrm{I}}$ and $\mathrm{Prob}_{\mathrm{II}}$ based on $\{y_t\}$, and the estimation results with different simulation lengths are summarized in Fig. 2(b). It can be seen that, in contrast to with the other methods, the spectral algorithm for equilibrium OOMs effectively reduce the statistical bias in the nonequilibrium data, and achieves statistically correct estimation at $T = 300$.

Figs. 2(c) and 2(d) plot estimates of stationary distribution of $\{x_t\}$ obtained from $\{x_t\}$ directly, where the empirical estimator calculates statistics through averaging over all observations. In this case, the proposed binless OOM significantly outperform the other methods, and its estimates are very close to true values even for extremely small short trajectories.

Fig. 3 provides an example of a two-dimensional diffusion process. The dynamics of this process can also be represented in the form of (24) and the potential function is shown in Fig. 3(a). The goal of this example is to estimate the first time-structure based independent component $w_{\mathrm{TICA}}$ [30] of this process from simulation data. Here $w_{\mathrm{TICA}}$ is a kinetic quantity of the process and is the solution to the generalized eigenvalue problem

$$C_\tau w = \lambda C_0 w$$

with the largest eigenvalue, where $C_0$ is the covariance matrix of $\{x_t\}$ in equilibrium and $C_\tau = \left( \mathbb{E}_\infty \left[ x_t x_{t+\tau}^\top \right] - \mathbb{E}_\infty [x_t] \mathbb{E}_\infty \left[ x_t^\top \right] \right)$ is the equilibrium time-lagged covariance matrix. The simulation data are also nonequilibrium with all simulations starting from the uniform distribution on $[-2, 0] \times [-2, 0]$. Fig. 3(b) displays the estimation errors of $w_{\mathrm{TICA}}$ obtained from different learning methods, which also demonstrates the superiority of the binless spectral method.

**Alanine dipeptide**  Alanine dipeptide is a small molecule which consists of two alanine amino acid units, and its configuration can be described by two backbone dihedral angles. Fig. 4(a) shows the potential profile of the alanine dipeptide with respect to the two angles, which contains five metastable

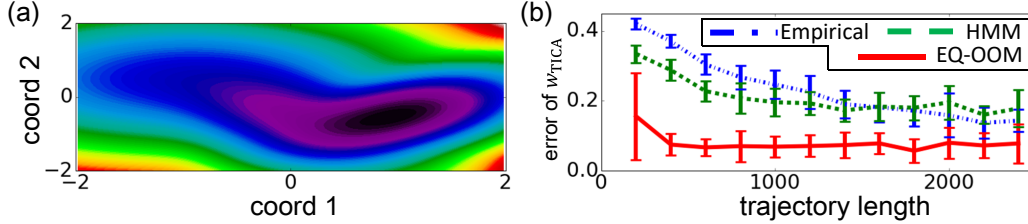

Figure 3: Comparison of modeling methods for a two-dimensional diffusion process. (a) Potential function. (b) Estimation error of $w_{\mathrm{TICA}} \in \mathbb{R}^2$ of the first TIC with lag time 100. Length of each trajectory is $T = 200 \sim 2500$ and the number of trajectories is $[10^5/T]$. Error bars are standard deviations over 30 independent experiments.

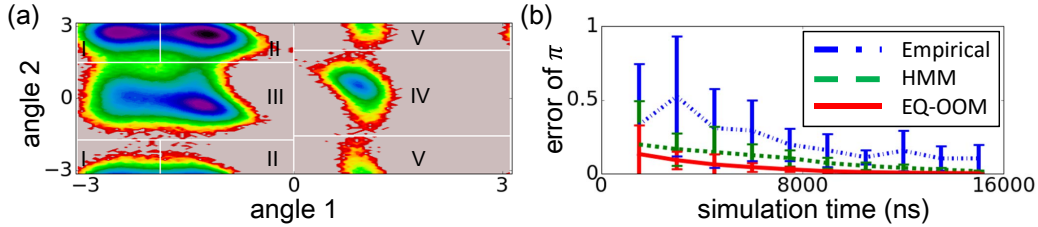

Figure 4: Comparison of modeling methods for molecular dynamics of alanine dipeptide. (a) Reduced free energy. (b) Estimation error of $\boldsymbol{\pi}$, where the horizontal axis denotes the total simulation time $T \times I$. Length of each trajectory is $T = 10\mathrm{ns}$ and the number of trajectories is $I = 150 \sim 1500$. Error bars are standard deviations over 30 independent experiments.

states $\{\mathrm{I}, \mathrm{II}, \mathrm{III}, \mathrm{IV}, \mathrm{V}\}$. We perform multiple short molecular dynamics simulations starting from the metastable state $\mathrm{IV}$, where each simulation length is $10\mathrm{ns}$, and utilizes different methods to approximate the stationary distribution $\pi = (\mathrm{Prob}_{\mathrm{I}}, \mathrm{Prob}_{\mathrm{II}}, \dots, \mathrm{Prob}_{\mathrm{V}})$ of the five metastable states. As shown in Fig. 4(b), the proposed binless algorithm yields lower estimation error compared to each of the alternatives.

## 6 Conclusion

In this paper, we investigated the statistical properties of the general spectral learning procedure for nonequilibrium data, and developed novel spectral methods for learning equilibrium dynamics from nonequilibrium (discrete or continuous) data. The main ideas of the presented methods are to correct the model parameters by the equilibrium constraint and to handle continuous observations in a binless manner. Interesting directions of future research include analysis of approximation error with finite data size and applications to controlled systems.

**Acknowledgments**

This work was funded by Deutsche Forschungsgemeinschaft (SFB 1114) and European Research Council (starting grant "pcCells").

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
