[Supplementary Material · si.pdf]

# Supplementary Information

## A   Proofs

### A.1   Proof of Theorem 1

For convenience, here we define

$$\boldsymbol{\omega}^*(T) = \frac{1}{T-2L}\boldsymbol{\omega}\sum_{t=1}^{T-2L}\boldsymbol{\Xi}(\mathcal{O})^{t-1}$$

$$\mathbf{G}_\sigma = \sum_{z_{1:L}}\boldsymbol{\phi}_2(z_{1:L})\boldsymbol{\sigma}^\top\boldsymbol{\Xi}(z_{1:L})^\top \tag{A.1}$$

**Part (1)**   We first show the theorem in the case of $T = T_0$ and $I \to \infty$.

Let

$$\mathbf{G}_\omega = \sum_{z_{1:L}}\boldsymbol{\phi}_1(z_{1:L})\boldsymbol{\omega}^*(T_0)\boldsymbol{\Xi}(z_{1:L}) \tag{A.2}$$

Since $I \to \infty$, we have

$$\hat{\bar{\boldsymbol{\phi}}}_1 \xrightarrow{p} \mathbb{E}\left[\hat{\bar{\boldsymbol{\phi}}}_1\right] = \mathbf{G}_\omega\boldsymbol{\sigma}$$

$$\hat{\bar{\boldsymbol{\phi}}}_2^\top \xrightarrow{p} \mathbb{E}\left[\hat{\bar{\boldsymbol{\phi}}}_2^\top\right] = \boldsymbol{\omega}^*(T_0)\mathbf{G}_\sigma^\top$$

$$\hat{\mathbf{C}}_{1,2} \xrightarrow{p} \mathbb{E}\left[\hat{\mathbf{C}}_{1,2}\right] = \mathbf{G}_\omega\mathbf{G}_\sigma^\top$$

$$\hat{\mathbf{C}}_{1,3}(x) \xrightarrow{p} \mathbb{E}\left[\hat{\mathbf{C}}_{1,2}(x)\right] = \mathbf{G}_\omega\boldsymbol{\Xi}(x)\mathbf{G}_\sigma^\top$$

According to Assumption 3 and the Eckart-Young-Mirsky Theorem, we can conclude that

$$\text{rank}(\mathbf{G}_\omega) = \text{rank}(\mathbf{G}_\sigma) = \text{rank}\left(\hat{\mathbf{C}}_{1,2}\right) = m$$

and

$$\hat{\mathbf{C}}_{1,2}^{\text{trun}} = \mathbf{U}\boldsymbol{\Sigma}\mathbf{V}^\top \xrightarrow{p} \mathbf{G}_\omega\mathbf{G}_\sigma^\top$$

By using the SVD of $\mathbf{G}_\omega\mathbf{G}_\sigma^\top$

$$\mathbf{G}_\omega\mathbf{G}_\sigma^\top = \tilde{\mathbf{U}}\tilde{\boldsymbol{\Sigma}}\tilde{\mathbf{V}}^\top$$

with $\text{rank}\left(\tilde{\mathbf{U}}\right) = \text{rank}\left(\tilde{\mathbf{V}}\right) = \text{rank}\left(\tilde{\boldsymbol{\Sigma}}\right)$, we can construct an OOM $\mathcal{M}' = (\boldsymbol{\omega}', \{\boldsymbol{\Xi}'(x)\}_{x\in\mathcal{O}}, \boldsymbol{\sigma}')$ with

$$\boldsymbol{\omega}' = \hat{\boldsymbol{\omega}}\left(\mathbf{G}_\sigma^\top\mathbf{V}\right)^{-1} \tag{A.3}$$

$$\boldsymbol{\Xi}'(x) = \left(\mathbf{G}_\sigma^\top\mathbf{V}\right)\hat{\boldsymbol{\Xi}}(x)\left(\mathbf{G}_\sigma^\top\mathbf{V}\right)^{-1} \tag{A.4}$$

$$\boldsymbol{\sigma}' = \left(\mathbf{G}_\sigma^\top\mathbf{V}\right)\hat{\boldsymbol{\sigma}} \tag{A.5}$$

which is obviously equivalent to $\hat{\mathcal{M}}$.

We can obtain from $\text{rank}\left(\mathbf{U}\boldsymbol{\Sigma}\mathbf{V}^\top\right) = \text{rank}\left(\mathbf{G}_\omega\mathbf{G}_\sigma^\top\right) = m$ that

$$\left(\mathbf{U}\boldsymbol{\Sigma}\mathbf{V}^\top\right)^+ = \mathbf{V}\boldsymbol{\Sigma}^{-1}\mathbf{U}^\top \xrightarrow{p} \left(\mathbf{G}_\omega\mathbf{G}_\sigma^\top\right)^+$$

where $\mathbf{A}^+$ denotes the Moore-Penrose pseudoinverse of $\mathbf{A}$, so

$$
\begin{aligned}
\boldsymbol{\omega}' &= \hat{\boldsymbol{\phi}}_2^\top \mathbf{V} \left(\mathbf{G}_\sigma^\top \mathbf{V}\right)^{-1} \\
&\xrightarrow{p} \boldsymbol{\omega}^*(T_0) \\
\boldsymbol{\Xi}'(x) &= \left(\mathbf{G}_\sigma^\top \mathbf{V}\right) \boldsymbol{\Sigma}^{-1} \mathbf{U}^\top \hat{\mathbf{C}}_{1,3}\left(x\right) \mathbf{V} \left(\mathbf{G}_\sigma^\top \mathbf{V}\right)^{-1} \\
&\xrightarrow{p} \mathbf{G}_\sigma^\top \mathbf{V} \boldsymbol{\Sigma}^{-1} \mathbf{U}^\top \mathbf{G}_\omega \boldsymbol{\Xi}(x) \\
&\xrightarrow{p} \mathbf{G}_\sigma^\top \left(\mathbf{G}_\omega \mathbf{G}_\sigma^\top\right)^+ \mathbf{G}_\omega \boldsymbol{\Xi}(x) \\
&= \mathbf{G}_\omega^+ \mathbf{G}_\omega \mathbf{G}_\sigma^\top \left(\mathbf{G}_\omega \mathbf{G}_\sigma^\top\right)^+ \mathbf{G}_\omega \mathbf{G}_\sigma^\top \mathbf{G}_\sigma^{+\top} \boldsymbol{\Xi}(x) \\
&= \boldsymbol{\Xi}(x) \\
\boldsymbol{\sigma}' &= \mathbf{G}_\sigma^\top \mathbf{V} \boldsymbol{\Sigma}^{-1} \mathbf{U}^\top \hat{\boldsymbol{\phi}}_1 \\
&\xrightarrow{p} \boldsymbol{\sigma}
\end{aligned}
$$

Note $\boldsymbol{\omega}' \xrightarrow{p} \boldsymbol{\omega}$ does not hold in general cases.

**Part (2)**   We now consider the case of $I = I_0$ and $T \to \infty$.

According to Assumption 2, the limit

$$
\begin{aligned}
\hat{\mathbf{C}}_{1,2} &\xrightarrow{p} \mathbb{E}_\infty \left[\boldsymbol{\phi}_1(x_{t-L:t-1})\boldsymbol{\phi}_2(x_{t:t+L-1})^\top\right] \\
&= \lim_{k \to \infty} \sum_{z_{1:L}} \boldsymbol{\phi}_1(z_{1:L}) \boldsymbol{\omega} \boldsymbol{\Xi}\left(\mathcal{O}\right)^k \boldsymbol{\Xi}(z_{1:L}) \mathbf{G}_\sigma^\top
\end{aligned}
$$

exists. Then

$$
\begin{aligned}
\hat{\boldsymbol{\phi}}_1 &\xrightarrow{p} \mathbb{E}_\infty \left[\boldsymbol{\phi}_1(x_{t-L:t-1})\right] = \mathbf{G}_\omega \boldsymbol{\sigma} \\
\hat{\boldsymbol{\phi}}_2^\top &\xrightarrow{p} \mathbb{E}_\infty \left[\boldsymbol{\phi}_2(x_{t:t+L-1})^\top\right] = \lim_{k \to \infty} \boldsymbol{\omega} \boldsymbol{\Xi}\left(\mathcal{O}\right)^k \mathbf{G}_\sigma^\top \\
\hat{\mathbf{C}}_{1,2} &\xrightarrow{p} \mathbb{E}_\infty \left[\hat{\mathbf{C}}_{1,2}\right] = \mathbf{G}_\omega \mathbf{G}_\sigma^\top \\
\hat{\mathbf{C}}_{1,3}\left(x\right) &\xrightarrow{p} \mathbb{E}_\infty \left[\hat{\mathbf{C}}_{1,2}\left(x\right)\right] = \mathbf{G}_\omega \boldsymbol{\Xi}(x) \mathbf{G}_\sigma^\top
\end{aligned}
$$

with

$$
\mathbf{G}_\omega = \lim_{k \to \infty} \sum_{z_{1:L}} \boldsymbol{\phi}_1(z_{1:L}) \boldsymbol{\omega} \boldsymbol{\Xi}\left(\mathcal{O}\right)^k \boldsymbol{\Xi}(z_{1:L}) \tag{A.6}
$$

The remaining part of the proof is omitted because it is the same as in Part (1).

## A.2   Asymptotic correctness of nonequilibrium learning with different initial states

If the $i$-th observation trajectories is generated by OOM $\mathcal{M} = (\boldsymbol{\omega}^i, \{\boldsymbol{\Xi}(x)\}_{x \in \mathcal{O}}, \boldsymbol{\sigma})$ for $i = 1, \dots, I$, and

$$
\boldsymbol{\omega}^{**} = \begin{cases} \frac{1}{I} \sum_{i=1}^I \boldsymbol{\omega}^i, & \text{for } T \to \infty \\ \mathrm{plim}_{I \to \infty} \frac{1}{I} \sum_{i=1}^I \boldsymbol{\omega}^i, & \text{for } I \to \infty \end{cases}
$$

the asymptotic correctness can also be shown as in Appendix A.1 by setting

$$
\mathbf{G}_\omega = \sum_{z_{1:L}} \boldsymbol{\phi}_1(z_{1:L}) \boldsymbol{\omega}^*(T_0) \boldsymbol{\Xi}(z_{1:L})
$$

with

$$
\boldsymbol{\omega}^*(T) = \frac{1}{T - 2L} \boldsymbol{\omega}^{**} \sum_{t=1}^{T-2L} \boldsymbol{\Xi}(\mathcal{O})^{t-1}
$$

for $I \to \infty$, and

$$
\mathbf{G}_\omega = \lim_{k \to \infty} \sum_{z_{1:L}} \boldsymbol{\phi}_1(z_{1:L}) \boldsymbol{\omega}^{**} \boldsymbol{\Xi}\left(\mathcal{O}\right)^k \boldsymbol{\Xi}(z_{1:L})
$$

for $T \to \infty$.

### A.3 Proof of Theorem 2

**Part (1)** We first show that there is an OOM $\mathcal{M}_{\text{eq}} = (\boldsymbol{\omega}_{\text{eq}}, \{\boldsymbol{\Xi}(x)\}_{x \in \mathcal{O}}, \boldsymbol{\sigma})$ which can describe the equilibrium dynamics of $\{x_t\}$.

In the case of $T = T_0$ and $I \to \infty$, we can obtain from Assumptions 2 and 3 that

$$
\begin{aligned}
\lim_{k \to \infty} \mathbf{G}_\omega \boldsymbol{\Xi}(\mathcal{O})^k \mathbf{G}_\sigma^\top &= \lim_{k \to \infty} \frac{1}{T_0 - 2L} \sum_{t=0}^{T_0 - 2L - 1} \mathbb{E}\left[ \boldsymbol{\phi}_1 \left( x_{t+1:t+L} \right) \boldsymbol{\phi}_2 \left( x_{t+L+k+1:t+2L+k} \right)^\top \right] \\
&= \left( \frac{1}{T_0 - 2L} \sum_{t=0}^{T_0 - 2L - 1} \mathbb{E}\left[ \boldsymbol{\phi}_1 \left( x_{t+1:t+L} \right) \right] \right) \left( \mathbb{E}_\infty \left[ \boldsymbol{\phi}_2 \left( x_{t+1:t+L} \right)^\top \right] \right) \\
&= \mathbf{G}_\omega \boldsymbol{\sigma} \left( \mathbb{E}_\infty \left[ \boldsymbol{\phi}_2 \left( x_{t+1:t+L} \right)^\top \right] \right) \\
\Rightarrow \lim_{k \to \infty} \boldsymbol{\Xi}(\mathcal{O})^k &= \boldsymbol{\sigma} \boldsymbol{\omega}_{\text{eq}}
\end{aligned}
\tag{A.7}
$$

with

$$
\boldsymbol{\omega}_{\text{eq}} = \left( \mathbb{E}_\infty \left[ \boldsymbol{\phi}_2 \left( x_{t+1:t+L} \right)^\top \right] \right) \mathbf{G}_\sigma^{+\top}
\tag{A.8}
$$

where $\mathbf{G}_\omega$ and $\mathbf{G}_\sigma$ are defined by (A.2) and (A.1). Then

$$
\begin{aligned}
\lim_{t \to \infty} \mathbb{P}\left( x_{t+1:t+l} = z_{1:l} \right) &= \lim_{t \to \infty} \boldsymbol{\omega} \boldsymbol{\Xi}(\mathcal{O})^t \boldsymbol{\Xi}(z_{1:l}) \boldsymbol{\sigma} \\
&= \boldsymbol{\omega} \boldsymbol{\Xi}(\mathcal{O}) \boldsymbol{\sigma} \boldsymbol{\omega}_{\text{eq}} \boldsymbol{\Xi}(z_{1:l}) \boldsymbol{\sigma} \\
&= \boldsymbol{\omega}_{\text{eq}} \boldsymbol{\Xi}(z_{1:l}) \boldsymbol{\sigma}
\end{aligned}
$$

In the case of $I = I_0$ and $T \to \infty$, because $\text{rank}\left( \mathbf{G}_\omega \right) = m$ for $\mathbf{G}_\omega$ defined by (A.6), there is a sufficiently large but finite $T'$ so that $\text{rank}\left( \mathbf{G}'_\omega \right) = m$ with

$$
\mathbf{G}'_\omega = \sum_{z_{1:L}} \boldsymbol{\phi}_1(z_{1:L}) \boldsymbol{\omega} \boldsymbol{\Xi}\left( \mathcal{O} \right)^{T'} \boldsymbol{\Xi}(z_{1:L})
$$

Considering

$$
\begin{aligned}
\lim_{k \to \infty} \mathbf{G}'_\omega \boldsymbol{\Xi}(\mathcal{O})^k \mathbf{G}_\sigma^\top &= \lim_{k \to \infty} \mathbb{E}\left[ \boldsymbol{\phi}_1 \left( x_{T'+1:T'+L} \right) \boldsymbol{\phi}_2 \left( x_{T'+L+k+1:T'+2L+k} \right)^\top \right] \\
&= \mathbf{G}'_\omega \boldsymbol{\sigma} \left( \mathbb{E}_\infty \left[ \boldsymbol{\phi}_2 \left( x_{t+1:t+L} \right)^\top \right] \right) \\
\Rightarrow \lim_{k \to \infty} \boldsymbol{\Xi}(\mathcal{O})^k &= \boldsymbol{\sigma} \boldsymbol{\omega}_{\text{eq}}
\end{aligned}
\tag{A.9}
$$

with $\boldsymbol{\omega}_{\text{eq}}$ defined by (A.8), we can also conclude that

$$
\lim_{t \to \infty} \mathbb{P}\left( x_{t+1:t+l} = z_{1:l} \right) = \boldsymbol{\omega}_{\text{eq}} \boldsymbol{\Xi}(z_{1:l}) \boldsymbol{\sigma}
$$

Note in both cases, $\boldsymbol{\omega}_{\text{eq}}$ satisfies $\boldsymbol{\omega}_{\text{eq}} \lim_{k \to \infty} \boldsymbol{\Xi}(\mathcal{O})^k = \boldsymbol{\omega}_{\text{eq}}$ and

$$
\begin{aligned}
\boldsymbol{\omega}_{\text{eq}} \boldsymbol{\Xi}(\mathcal{O}) &= \lim_{t \to \infty} \boldsymbol{\omega}_{\text{eq}} \boldsymbol{\Xi}(\mathcal{O})^{t+1} \\
&= \boldsymbol{\omega}_{\text{eq}} \\
\boldsymbol{\omega}_{\text{eq}} \boldsymbol{\sigma} &= \boldsymbol{\omega}_{\text{eq}} \boldsymbol{\Xi}(\mathcal{O}) \boldsymbol{\sigma} \\
&= \lim_{t \to \infty} \sum_{x \in \mathcal{O}} \mathbb{P}\left( x_t = x \right) = 1
\end{aligned}
$$

**Part (2)** In this part, we show that

$$
\mathbf{w} \boldsymbol{\Xi}(\mathcal{O}) = \mathbf{w}, \quad \mathbf{w} \boldsymbol{\sigma} = 1
$$

has a unique solution $\mathbf{w} = \boldsymbol{\omega}_{\text{eq}}$.

According to Appendix A.1 and (A.7), (A.9), if $\mathbf{w} \boldsymbol{\Xi}(\mathcal{O}) = \mathbf{w}$ and $\mathbf{w} \boldsymbol{\sigma} = 1$, we have

$$
\begin{aligned}
\mathbf{w} &= \lim_{k \to \infty} \mathbf{w} \boldsymbol{\Xi}(\mathcal{O})^k \\
&= \mathbf{w} \boldsymbol{\sigma} \boldsymbol{\omega}_{\text{eq}} \\
&= \boldsymbol{\omega}_{\text{eq}}
\end{aligned}
$$

**Part (3)**  We now show Theorem 2.

The problem (16) is equivalent to

$$
\begin{aligned}
\min_{\mathbf{w}'} E\left(\mathbf{w}'\right) \;=\;& \left(\mathbf{w}'\boldsymbol{\Xi}'\left(\mathcal{O}\right)-\mathbf{w}'\right)\left(\mathbf{G}_\sigma^\top\mathbf{V}\right) \\
& \cdot\left(\mathbf{G}_\sigma^\top\mathbf{V}\right)^\top\left(\mathbf{w}'\boldsymbol{\Xi}'\left(\mathcal{O}\right)-\mathbf{w}'\right)^\top \\
\text{s.t.} \qquad & \mathbf{w}'\boldsymbol{\sigma}'=1
\end{aligned}
$$

where $\boldsymbol{\Xi}'\left(\mathcal{O}\right)=\sum_{x\in\mathcal{O}}\boldsymbol{\Xi}'\left(x\right)$, $\boldsymbol{\Xi}'\left(x\right)$ and $\boldsymbol{\sigma}'$ are given by (A.4) and (A.5), and $\mathbf{w}'$ is related to $\mathbf{w}$ with $\mathbf{w}'=\mathbf{w}\left(\mathbf{G}_\sigma^\top\mathbf{V}\right)^{-1}$. This problem can be further transformed into an unconstrained one

$$
\min_{\mathbf{w}'} E\left(\mathbf{w}'\left(\mathbf{I}-\boldsymbol{\sigma}'\boldsymbol{\sigma}'^{+}\right)+\boldsymbol{\sigma}'^{+}\right)+\left\|\mathbf{w}'\left(\mathbf{I}-\boldsymbol{\sigma}'\boldsymbol{\sigma}'^{+}\right)+\boldsymbol{\sigma}'^{+}-\mathbf{w}'\right\|^2 \tag{A.10}
$$

where $\mathbf{w}'\left(\mathbf{I}-\boldsymbol{\sigma}'\boldsymbol{\sigma}'^{+}\right)+\boldsymbol{\sigma}'^{+}$ is the projection of $\mathbf{w}'$ on the space $\{\mathbf{w}'|\mathbf{w}'\boldsymbol{\sigma}'=1\}$ and $\mathbf{I}$ denotes the identity matrix of appropriate dimension. Considering that $\boldsymbol{\Xi}'\left(x\right)\xrightarrow{P}\boldsymbol{\Xi}\left(x\right)$, $\boldsymbol{\sigma}'\xrightarrow{P}\boldsymbol{\sigma}$,

$$
\begin{aligned}
\left(\mathbf{G}_\sigma^\top\mathbf{V}\right)\left(\mathbf{G}_\sigma^\top\mathbf{V}\right)^\top \;=\;& \mathbf{G}_\sigma^\top\mathbf{V}\boldsymbol{\Sigma}^{-1}\mathbf{U}^\top\mathbf{U}\boldsymbol{\Sigma}\mathbf{V}^\top\mathbf{G}_\sigma \\
\xrightarrow{P}\;& \mathbf{G}_\sigma^\top\left(\mathbf{G}_\omega^\top\mathbf{G}_\sigma\right)^{+}\mathbf{G}_\omega\mathbf{G}_\sigma^\top\mathbf{G}_\sigma \\
=\;& \mathbf{G}_\sigma^\top\mathbf{G}_\sigma
\end{aligned}
$$

and the conclusion in Part (2), we can obtain that the optimal solution of (A.10) converges to $\boldsymbol{\omega}_{\mathrm{eq}}$ in probability and $\hat{\boldsymbol{\omega}}_{\mathrm{eq}}\xrightarrow{P}\boldsymbol{\omega}_{\mathrm{eq}}\left(\mathbf{G}_\sigma^\top\mathbf{V}\right)^{-1}$ according to Theorem 2.7 in [1], which yields the conclusion of Theorem 2.

**Part (4)**  We derive in this part the closed-form solution to (16).

Since the projection of $\mathbf{w}''$ on the space $\{\mathbf{w}''|\mathbf{w}''\hat{\boldsymbol{\sigma}}=1\}$ is $\mathbf{w}''\left(\mathbf{I}-\hat{\boldsymbol{\sigma}}\hat{\boldsymbol{\sigma}}^{+}\right)+\hat{\boldsymbol{\sigma}}^{+}$, (16) can be equivalent transformed into

$$
\min_{\mathbf{w}''}\left\|\mathbf{w}''\left(\mathbf{I}-\hat{\boldsymbol{\sigma}}\hat{\boldsymbol{\sigma}}^{+}\right)\left(\hat{\boldsymbol{\Xi}}(\mathcal{O})-\mathbf{I}\right)+\hat{\boldsymbol{\sigma}}^{+}\left(\hat{\boldsymbol{\Xi}}(\mathcal{O})-\mathbf{I}\right)\right\|^2
$$

The solution to this problem is

$$
\mathbf{w}^{*}=-\hat{\boldsymbol{\sigma}}^{+}\left(\hat{\boldsymbol{\Xi}}(\mathcal{O})-\mathbf{I}\right)\left(\left(\mathbf{I}-\hat{\boldsymbol{\sigma}}\hat{\boldsymbol{\sigma}}^{+}\right)\left(\hat{\boldsymbol{\Xi}}(\mathcal{O})-\mathbf{I}\right)\right)^{+}
$$

which provides the optimal value of $\hat{\boldsymbol{\omega}}_{\mathrm{eq}}$ as

$$
\begin{aligned}
\hat{\boldsymbol{\omega}}_{\mathrm{eq}} \;=\;& \mathbf{w}^{*}\left(\mathbf{I}-\hat{\boldsymbol{\sigma}}\hat{\boldsymbol{\sigma}}^{+}\right)+\hat{\boldsymbol{\sigma}}^{+} \\
=\;& \hat{\boldsymbol{\sigma}}^{+}-\hat{\boldsymbol{\sigma}}^{+}\left(\hat{\boldsymbol{\Xi}}(\mathcal{O})-\mathbf{I}\right)\left(\left(\mathbf{I}-\hat{\boldsymbol{\sigma}}\hat{\boldsymbol{\sigma}}^{+}\right)\left(\hat{\boldsymbol{\Xi}}(\mathcal{O})-\mathbf{I}\right)\right)^{+}\left(\mathbf{I}-\hat{\boldsymbol{\sigma}}\hat{\boldsymbol{\sigma}}^{+}\right) \tag{A.11}
\end{aligned}
$$

## A.4  Proof of Theorem 3

Here we only consider the consistency of the binless OOM as $I\to\infty$. The proof can be easily to extended to the case of $T\to\infty$. In addition, we denote $\mathbb{E}_\infty[g(x_{t+1:t+r})]$ and $\mathbb{E}[g(x_{1:r})|\hat{\mathcal{M}}_{\mathrm{eq}}]$ by $\mathbb{E}_\infty[g]$ and $\mathbb{E}_{\hat{\mathcal{M}}}[g]$ for convenience of notation.

**Part (1)**  We first show that Theorem 3 holds for $g\left(x_{t+1:t+r}\right)=1_{x_{t+1:t+r}\in\mathcal{B}_{i_1}\times\mathcal{B}_{i_2}\times\ldots\times\mathcal{B}_{i_r}}$, where $\mathcal{B}_1,\ldots,\mathcal{B}_K$ is a partition of $\mathcal{O}$ and $i_{1:r}\in\{1,\ldots,K\}^r$. In this case, we can construct a discrete OOM with observation space $\{\mathcal{B}_1,\ldots,\mathcal{B}_K\}$ by the nonequilibrium learning algorithm, which can provide the same estimate of $\mathbb{E}_\infty\left[g\left(x_{t+1:t+r}\right)\right]$ as $\hat{\mathcal{M}}_{\mathrm{eq}}$. Therefore, we can show $\mathbb{E}_{\hat{\mathcal{M}}}[g]\xrightarrow{P}\mathbb{E}_\infty[g]$ by using the similar proof of Theorem 2.

**Part (2)**  We now consider the case that $g$ is a continuous function. According to the Heine-Cantor theorem, $g$ is also uniformly continuous. Then, for an arbitrary $\epsilon>0$, we can construct a simple function

$$
\hat{g}(x_{t+1:t+r})=\sum_{i_1,\ldots,i_r}c_{i_1 i_2\ldots i_r}1_{x_{t+1:t+r}\in\mathcal{B}_{i_1}\times\ldots\times\mathcal{B}_{i_r}}
$$

so that
$$|g(z_{1:r}) - \hat{g}(z_{1:r})| \le \epsilon, \quad \forall z_{1:r} \in \mathcal{O}^r$$
where $\{\mathcal{B}_1, \dots, \mathcal{B}_K\}$ is a partition of $\mathcal{O}$. Then, we have
$$|\mathbb{E}_\infty[g] - \mathbb{E}_\infty[\hat{g}]| \le \mathbb{E}_\infty[|g - \hat{g}|] \le \epsilon$$
and
$$\left| \mathbb{E}_\infty[\hat{g}] - \mathbb{E}_{\hat{\mathcal{M}}}[\hat{g}] \right| \xrightarrow{p} 0$$
as $I \to \infty$ according to the conclusion of Part (1), where $\mathbb{E}_\infty[g] = \mathbb{E}_\infty[g(x_{t+1:t+r})]$ and $\mathbb{E}_{\hat{\mathcal{M}}}[g] = \mathbb{E}[g(x_{1:r})|\hat{\mathcal{M}}_{\text{eq}}]$.

It can be known from the boundness of feature functions, there exists a constant $\xi$ such that
$$1_{\max_{x \in \mathcal{X}} \|\hat{\mathbf{W}}_x\| < \xi / |\mathcal{X}|} \xrightarrow{p} 1 \tag{A.12}$$
Under the condition that $\max_{x \in \mathcal{X}} \left\| \hat{\mathbf{W}}_x \right\| < \xi / |\mathcal{X}|$, we have

$$
\begin{aligned}
\left| \mathbb{E}_{\hat{\mathcal{M}}_{\text{eq}}}[\hat{g}] - \mathbb{E}_{\hat{\mathcal{M}}_{\text{eq}}}[g] \right| &= \hat{\boldsymbol{\omega}}_{\text{eq}} \left( \sum_{z_{1:r} \in \mathcal{X}^r} (\hat{g}(z_{1:r}) - g(z_{1:r})) \mathbf{W}_{z_1} \dots \mathbf{W}_{z_r} \right) \hat{\boldsymbol{\sigma}} \\
&\le \|\hat{\boldsymbol{\omega}}_{\text{eq}}\| \|\hat{\boldsymbol{\sigma}}\| \left( \sum_{z_{1:r} \in \mathcal{X}^r} \frac{\xi^r \epsilon}{|\mathcal{X}|^r} \right) \\
&= \|\hat{\boldsymbol{\omega}}_{\text{eq}}\| \|\hat{\boldsymbol{\sigma}}\| \xi^r \epsilon
\end{aligned}
$$

In addition, considering that we can show as in Appendix A.1 that
$$
\begin{aligned}
\hat{\boldsymbol{\omega}}_{\text{eq}} &\xrightarrow{p} \boldsymbol{\omega}_{\text{eq}} \mathbf{G}_\sigma^\top \mathbf{V} \\
\hat{\boldsymbol{\sigma}} &\xrightarrow{p} \left( \mathbf{G}_\sigma^\top \mathbf{V} \right)^{-1} \boldsymbol{\sigma}
\end{aligned}
$$
we can obtain
$$1_{\|\hat{\boldsymbol{\omega}}_{\text{eq}}\| \|\hat{\boldsymbol{\sigma}}\| \le \xi_0} \xrightarrow{p} 1 \tag{A.13}$$
and
$$1_{\left| \mathbb{E}_{\hat{\mathcal{M}}_{\text{eq}}}[\hat{g}] - \mathbb{E}_{\hat{\mathcal{M}}_{\text{eq}}}[g] \right| \le \xi_0 \xi^r \epsilon} \xrightarrow{p} 1$$
where $\xi_0$ is a constant larger than $\|\hat{\boldsymbol{\omega}}_{\text{eq}}\| \cdot \|\hat{\boldsymbol{\sigma}}\|$.

Based on the above analysis and the fact that
$$
\begin{aligned}
\left| \mathbb{E}_\infty[g] - \mathbb{E}_{\hat{\mathcal{M}}_{\text{eq}}}[g] \right| &= \left| \mathbb{E}_\infty[g] - \mathbb{E}_\infty[\hat{g}] + \mathbb{E}_\infty[\hat{g}] - \mathbb{E}_{\hat{\mathcal{M}}_{\text{eq}}}[\hat{g}] + \mathbb{E}_{\hat{\mathcal{M}}_{\text{eq}}}[\hat{g}] - \mathbb{E}_{\hat{\mathcal{M}}_{\text{eq}}}[g] \right| \\
&\le |\mathbb{E}_\infty[g] - \mathbb{E}_\infty[\hat{g}]| + \left| \mathbb{E}_\infty[\hat{g}] - \mathbb{E}_{\hat{\mathcal{M}}_{\text{eq}}}[\hat{g}] \right| + \left| \mathbb{E}_{\hat{\mathcal{M}}_{\text{eq}}}[\hat{g}] - \mathbb{E}_{\hat{\mathcal{M}}_{\text{eq}}}[g] \right|
\end{aligned}
$$
we can get
$$
\begin{aligned}
\Pr\left( \left| \mathbb{E}_\infty[g] - \mathbb{E}_{\hat{\mathcal{M}}_{\text{eq}}}[g] \right| \le (\xi_0 \xi^r + 2) \epsilon \right) &\ge \Pr\left( |\mathbb{E}_\infty[g] - \mathbb{E}_\infty[\hat{g}]| \le \epsilon, \left| \mathbb{E}_\infty[\hat{g}] - \mathbb{E}_{\hat{\mathcal{M}}_{\text{eq}}}[\hat{g}] \right| \le \epsilon, \right. \\
&\qquad\qquad \left. \left| \mathbb{E}_{\hat{\mathcal{M}}_{\text{eq}}}[\hat{g}] - \mathbb{E}_{\hat{\mathcal{M}}_{\text{eq}}}[g] \right| \le \xi_0 \xi^r \epsilon \right) \\
&\to 1
\end{aligned}
$$
Because this equation holds for all $\epsilon > 0$, we can conclude that $\mathbb{E}_{\hat{\mathcal{M}}_{\text{eq}}}[g] \xrightarrow{p} \mathbb{E}_\infty[g]$.

**Part (3)** In this part, we prove the conclusion of the theorem in the case where $g$ is a Borel measurable function and bounded with $|g(z_{1:r})| < \xi_g$ for all $z_{1:r} \in \mathcal{O}^r$, and there exist constants $\bar{\xi}$ and $\underline{\xi}$ so that $\|\Xi(x)\| \le \bar{\xi}$ and $\lim_{t \to \infty} \mathbb{P}(x_{t+1:t+r} = z_{1:r}) \ge \underline{\xi}$ for all $x \in \mathcal{O}$ and $z_{1:r} \in \mathcal{O}^r$.

According to Theorem 2.2 in [2], for an arbitrary $\epsilon > 0$, there is a continuous function $\hat{g}'$ satisfies $\mathbb{E}_\infty[1_{x_{t+1:t+r} \in \mathcal{K}_\epsilon(\hat{g}')}] < \epsilon$, where $\mathcal{K}_\epsilon(\hat{g}') = \{z_{1:r} | z_{1:r} \in \mathcal{O}^r, |\hat{g}'(z_{1:r}) - g(z_{1:r})| > \epsilon\}$. Define
$$
\hat{g}(z_{1:r}) = \begin{cases} \hat{g}'(z_{1:r}), & |\hat{g}'(z_{1:r})| \le \xi_g \\ -\xi_g, & \hat{g}'(z_{1:r}) < -\xi_g \\ \xi_g, & \hat{g}'(z_{1:r}) > \xi_g \end{cases}
$$

It can be seen that $\hat{g}$ is a continuous function which is also satisfies $\mathbb{E}_\infty[1_{x_{t+1:t+r}\in\mathcal{K}_\epsilon(\hat{g})}] < \epsilon$ and bounded with $|\hat{g}(z_{1:r})| < \xi_g$. So the difference between $\mathbb{E}_\infty[g]$ and $\mathbb{E}_\infty[\hat{g}]$ satisfies

$$
\begin{aligned}
|\mathbb{E}_\infty[g] - \mathbb{E}_\infty[\hat{g}]| &\leq \mathbb{E}_\infty\left[|g(x_{t+1:t+r}) - \hat{g}(x_{t+1:t+r})|\right] \\
&= \mathbb{E}_\infty[1_{x_{t+1:t+r}\in\mathcal{K}_\epsilon(\hat{g})}]\mathbb{E}_\infty\left[|g(x_{t+1:t+r}) - \hat{g}(x_{t+1:t+r})|\,|\,x_{t+1:t+r}\in\mathcal{K}_\epsilon(\hat{g})\right] \\
&\quad + \mathbb{E}_\infty[1_{x_{t+1:t+r}\notin\mathcal{K}_\epsilon(\hat{g})}]\mathbb{E}_\infty\left[|g(x_{t+1:t+r}) - \hat{g}(x_{t+1:t+r})|\,|\,x_{t+1:t+r}\notin\mathcal{K}_\epsilon(\hat{g})\right] \\
&\leq \epsilon\cdot 2\xi_g + \epsilon = (2\xi_g + 1)\,\epsilon
\end{aligned}
$$

For the difference between $\mathbb{E}_\infty[\hat{g}]$ and $\mathbb{E}_{\hat{\mathcal{M}}_{\text{eq}}}[\hat{g}]$, we can obtain from the above that $\left|\mathbb{E}_\infty[\hat{g}] - \mathbb{E}_{\hat{\mathcal{M}}_{\text{eq}}}[\hat{g}]\right| \xrightarrow{p} 0$ as $I \to \infty$ by considering that $\hat{g}$ is continuous, which implies that there is an $I_0$ such that

$$
\Pr\left(\left|\mathbb{E}_\infty[\hat{g}] - \mathbb{E}_{\hat{\mathcal{M}}_{\text{eq}}}[\hat{g}]\right| > \epsilon\right) < \epsilon, \quad \forall I > I_0
$$

Next, let us consider the value of $\left|\mathbb{E}_{\hat{\mathcal{M}}_{\text{eq}}}[\hat{g}] - \mathbb{E}_{\hat{\mathcal{M}}_{\text{eq}}}[g]\right|$. Note that

$$
\begin{aligned}
\left|\mathbb{E}_{\hat{\mathcal{M}}}[\hat{g}] - \mathbb{E}_{\hat{\mathcal{M}}}[g]\right| &\leq \|\hat{\boldsymbol{\omega}}_0\|\,\|\hat{\boldsymbol{\sigma}}\|\left\|\sum_{z_{1:n}\in\mathcal{X}^r}(\hat{g}(z_{1:r}) - g(z_{1:r}))\,\hat{\mathbf{W}}_{z_1}\dots\hat{\mathbf{W}}_{z_r}\right\| \\
&< \frac{\xi_0\xi^r}{|\mathcal{X}|^r}\left|\sum_{z_{1:r}\in\mathcal{X}^r}(\hat{g}(z_{1:r}) - g(z_{1:r}))\right|
\end{aligned}
$$

under the condition that $\left\|\hat{\mathbf{W}}_x\right\| < \xi/|\mathcal{X}|$ and $\|\hat{\boldsymbol{\omega}}_{\text{eq}}\|\,\|\hat{\boldsymbol{\sigma}}\| \leq \xi_0$. Therefore, there exists an $I_1$ such that

$$
\Pr\left(\left|\mathbb{E}_{\hat{\mathcal{M}}_{\text{eq}}}[\hat{g}] - \mathbb{E}_{\hat{\mathcal{M}}_{\text{eq}}}[g]\right| \geq \frac{\xi_0\xi^r}{|\mathcal{X}|^r}\left|\sum_{z_{1:r}\in\mathcal{X}^r}(\hat{g}(z_{1:r}) - g(z_{1:r}))\right|\right) < \epsilon, \quad \forall I > I_1 \tag{A.14}
$$

due to (A.12) and (A.13). Let $x'_{1:r}$ denotes a random sample taken uniformly from $\mathcal{X}^r$. We can obtain that

$$
\begin{aligned}
\mathbb{P}\left(x'_{1:r}\right) &= \mathbb{P}\left(x'_1\right)\dots\mathbb{P}\left(x'_r\right) \\
&\leq \left(\|\boldsymbol{\omega}\|\,\|\boldsymbol{\sigma}\|\,\xi_O\bar{\xi}\right)^r
\end{aligned}
$$

where $\xi_O \geq \left\|\boldsymbol{\Xi}\left(\mathcal{O}\right)^k\right\|$ for any $k \geq 0$. Note $\xi_O < \infty$ because we can show the existing of the limit of $\{\left\|\boldsymbol{\Xi}\left(\mathcal{O}\right)^0\right\|, \left\|\boldsymbol{\Xi}\left(\mathcal{O}\right)^1\right\|, \dots\}$ by similar steps in Appendix A.3. Thus

$$
\begin{aligned}
\mathbb{E}\left[\frac{1}{|\mathcal{X}|^r}\left|\sum_{z_{1:r}\in\mathcal{X}^r}(\hat{g}(z_{1:r}) - g(z_{1:r}))\right|\right] &\leq \mathbb{E}\left[\mathbb{E}\left[|\hat{g}(x'_{1:r}) - g(x'_{1:r})|\,|\,\mathcal{X}\right]\right] \\
&= \mathbb{E}\left[|\hat{g}(x'_{1:r}) - g(x'_{1:r})|\right] \\
&= \mathbb{E}\left[1_{x'_{1:r}\in\mathcal{K}_\epsilon(\hat{g})}\right]\mathbb{E}\left[|\hat{g}(x'_{1:r}) - g(x'_{1:r})|\,|\,x'_{1:r}\in\mathcal{K}_\epsilon(\hat{g})\right] \\
&\quad + \mathbb{E}\left[1_{x'_{1:r}\notin\mathcal{K}_\epsilon(\hat{g})}\right]\mathbb{E}\left[|\hat{g}(x'_{1:r}) - g(x'_{1:r})|\,|\,x'_{1:r}\notin\mathcal{K}_\epsilon(\hat{g})\right] \\
&\leq \xi_\mu\epsilon\cdot 2\xi_g + \epsilon = (2\xi_g\xi_\mu + 1)\,\epsilon
\end{aligned}
$$

where $\xi_\mu = \left(\|\boldsymbol{\omega}\|\,\|\boldsymbol{\sigma}\|\,\xi_O\bar{\xi}\right)^r/\underline{\xi}$. By the Markov's inequality, we have

$$
\Pr\left[\frac{1}{|\mathcal{X}|^r}\left|\sum_{z_{1:r}\in\mathcal{X}^r}(\hat{g}(z_{1:r}) - g(z_{1:r}))\right| \geq \sqrt{\epsilon}\right] \leq (2\xi_g\xi_\mu + 1)\sqrt{\epsilon} \tag{A.15}
$$

Combining (A.14) and (A.15) leads to

$$
\begin{aligned}
\Pr\left(\left|\mathbb{E}_{\hat{\mathcal{M}}_{\text{eq}}}[\hat{g}] - \mathbb{E}_{\hat{\mathcal{M}}_{\text{eq}}}[g]\right| \geq \xi_0\xi^r\sqrt{\epsilon}\right) &\leq \Pr\left(\left|\mathbb{E}_{\hat{\mathcal{M}}_{\text{eq}}}[\hat{g}] - \mathbb{E}_{\hat{\mathcal{M}}_{\text{eq}}}[g]\right| \geq \frac{\xi_0\xi^r}{|\mathcal{X}|^r}\left|\sum_{z_{1:r}\in X^r}(\hat{g}(z_{1:r}) - g(z_{1:r}))\right|\right) \\
&\quad + \Pr\left(\frac{1}{|\mathcal{X}|^r}\left|\sum_{z_{1:r}\in\mathcal{X}^r}(\hat{g}(z_{1:r}) - g(z_{1:r}))\right| \geq \sqrt{\epsilon}\right) \\
&\leq \epsilon + (2\xi_g\xi_\mu + 1)\sqrt{\epsilon}
\end{aligned}
$$

for all $I > I_1$.

From all the above, we have

$$\Pr\left(\left|\mathbb{E}_\infty[g] - \mathbb{E}_{\hat{\mathcal{M}}_{\text{eq}}}[g]\right| \le 2(\xi_g + 1)\epsilon + \xi_0 \xi^r \sqrt{\epsilon}\right)$$

$$\ge \Pr\left(\left|\mathbb{E}_\infty[\hat{g}] - \mathbb{E}_{\hat{\mathcal{M}}_{\text{eq}}}[\hat{g}]\right| \le \epsilon, \left|\mathbb{E}_{\hat{\mathcal{M}}_{\text{eq}}}[\hat{g}] - \mathbb{E}_{\hat{\mathcal{M}}_{\text{eq}}}[g]\right| \le \xi_0 \xi^r \sqrt{\epsilon}\right)$$

$$\ge 1 - \Pr\left(\left|\mathbb{E}_\infty[\hat{g}] - \mathbb{E}_{\hat{\mathcal{M}}_{\text{eq}}}[\hat{g}]\right| > \epsilon\right) - \Pr\left(\left|\mathbb{E}_{\hat{\mathcal{M}}_{\text{eq}}}[\hat{g}] - \mathbb{E}_{\hat{\mathcal{M}}_{\text{eq}}}[g]\right| > \xi_0 \xi^r \sqrt{\epsilon}\right)$$

$$\ge 1 - 2\epsilon - (2\xi_g \xi_\mu + 1)\sqrt{\epsilon}$$

for all $I > \max\{I_0, I_1\}$, which yields $\mathbb{E}_{\hat{\mathcal{M}}_{\text{eq}}}[g] \xrightarrow{p} \mathbb{E}_\infty[g]$ due to the arbitrariness of $\epsilon$.

# B  Settings in applications

## B.1  Models

The one-dimensional diffusion processes in Section 5 are driven by the Brownian dynamics with $\beta = 0.3$,

$$V(x) = \frac{\sum_{i=1}^{5}\left(|x - c_i| + 0.001\right)^{-2} u_i}{\sum_{i=1}^{5}\left(|x - c_i| + 0.001\right)^{-2}}$$

and the sample interval is 0.002. For the two-dimensional process, $\beta = 2$,

$$V(x) = -\log\left(\sum_{i=1}^{3} p_i \mathcal{N}(x|\mu_i, \Sigma_i)\right)$$

and the sample interval is 0.01, where $c_{1:5} = (-0.3, 0.5, 1, 1.5, 2.3)$, $u_{1:5} = (21, 4, 8, -1, 20)$, $p_{1:3} = (0.25, 0.25, 0.5)$, $\mu_1 = (0, -0.5)$, $\mu_2 = (-1, 0.5)$, $\mu_3 = (1, -0.5)$. The simulation details of alanine dipeptide is given in [3].

## B.2  Algorithms

The parameters of discrete spectral learning are chosen as: $L = 3$, $m = 10$, and $\phi_1 = \phi_2$ are indicator functions of all $\mathcal{O}^L$ observation subsequences with length $L$.

The parameters of binless spectral learning are almost the same as discrete ones, except $\phi_1 = \phi_2$ are Gaussian activation functions with random weights of functional link neural networks with $D_1 = D_2 = 100$.

The number of hidden states of HMMs is 10. For continuous data, we partition the state space into 100 discrete bins $k$-mean clustering, and then learn HMMs by the EM algorithm, where the HMM package in PyEMMA [4] is used. All observation samples within the same bin are assumed to be independent for quantitative analysis.