[Reviews · NeurIPS 2016]

Reviewer 1

Summary

The paper presents a spectral learning method for learning dynamical observable operator models in finite observation spaces from non-equilibrium data.

Qualitative Assessment

Update after author feedback: I thank the authors for their constructive response and hope they can incorporate as many of the promised changes as possible. Based on the feedback and reviewer discussion I now have a better idea of the motivation behind the work. Perhaps a brief description of a concrete example application problem motivating the work would make it easier for others more used to machine learning type learning of dynamical models to appreciate the work too. ---------------------------- Technical quality: The method looks interesting, but its foundation feels shaky as it is not clear that the presented probability models (1) and (2) are valid probability models. The experimental evaluation would benefit a lot from explicit comparisons with Bayesian alternatives (e.g. Ruttor et al., NIPS 2013; Svensson et al., AISTATS 2016 and references therein) to properly understand the pros and cons of the different approaches. Novelty: The paper presents a few novel extensions to existing work that clearly broaden its applicability. Impact: Based on the presentation, it is difficult to see how the paper could have a significant impact. The model applied is non-standard, but the authors do not really show how to use it for the typical problems in time series like prediction. The real examples seem to be more about inference but no advice is provided how and when the method could/should be applied to such problems. The paper would likely only be of interest to the small community who are already familiar with the methods. Clarity: At least for someone not familiar with the OOM methods before, the paper was really difficult to read as the foundations of the methods were not presented. A practical example might help, although it could be difficult to fit to the NIPS page limit. The experiments were presented too briefly to understand what was actually done (what was the data, how it was used, what was inferred and how that was obtained from the model?). Other comments: 1. Please clarify why Eqs. (1) and (2) define proper probability models. Based on the definition it is not clear that they would be positive or properly normalised. 2. Algorithm 1 is not really an algorithm as you do not specify how to select F_1 and F_2. 3. Line 189: Part of the theorem seems redundant as probabilities are non-negative making the lower bound trivial. Typo: Line 149: "quandratic"

Confidence in this Review

1-Less confident (might not have understood significant parts)


Reviewer 2

Summary

This study addresses the problem of moment based learning in Observable Operators Models when observable data is not necessarily in equilibrium state. The main contribution of this work consists in showing that standard moment based estimation of OOM lead to asymptotically consistent estimation of the model parameters with respect to either the number of observations or the length of the trajectories. The authors further extend the proposed theory to the continuous case, by showing the consistency of moment based estimation when considering continuous feature functions. The experimental results show that the proposed approach outperforms traditional OOM and expectation maximisation based HMM in estimating equilibrium probabilities when applied to both discrete and continuous observations.

Qualitative Assessment

The topic addressed in this work falls outside my field of expertise. Nevertheless, I found the paper rather clearly written and rigorous in the formulation. I wonder whether the assumptions underlying the proposed theory are easily satisfied in real case scenarios, and I would recommend to the authors to further discuss the practical implications, for example related to eventual numerical issues associated to Assumption 3.

Confidence in this Review

1-Less confident (might not have understood significant parts)


Reviewer 3

Summary

This paper presents a learning algorithm for Observable Operator Models (OOMs) that differs from existing approaches in that it does not assume the training data samples a system in equilibrium. This innovation should make OOMs applicable for modeling stochastic systems that are still changing. One example of such a system that was mentioned is protein folding. This paper also proposes a binless approach that makes OOMs applicable for systems involving continuous values.

Qualitative Assessment

OOMs are a quite recent approach for modeling stochastic processes. Despite their newness, this paper makes no effort to present this work in a manner that would be accessible to someone not already very familiar with OOMs. Consequently, this paper would be much better suited for a venue dedicated to stochastic processes. As far as I could determine, the contributions presented in this paper represent significant steps for OOMs, and the descriptions of these algorithms appear to be thorough. However, the validation is less convincing--it shows that each of the new methods is able to significantly outperform HMMs at modeling a sequence of generated data. Statistical analysis suggests that the new methods performed very well, but demonstrations with real-world data were lacking. The results were not interpreted in a manner that I could understand.

Confidence in this Review

1-Less confident (might not have understood significant parts)


Reviewer 4

Summary

The authors present a nonequilibrium learning algorithm for OOM.

Qualitative Assessment

I am not an expert of OOM and for this reason to me it is difficult to follow the derivations. However, I think that the presentation of the paper should be improved to enhance the clarity.

Confidence in this Review

1-Less confident (might not have understood significant parts)


Reviewer 5

Summary

This paper concerns the modeling of discrete- and continuous-valued stochastic processes with observable operator models (OOMs), which (together with a number of closely related models like predictive state representations and multiplicity automata) offer fast constructive and asymptotically correct learning algorithms for a model class that properly include HMMs. Specifically, this article focusses on nonstationary processes. Technical contributions: 1. proofs that the standard model estimation techniques generalize immediately from stationary to nonstationary data (Section 3). 2. An algorithm to reconstruct (an estimate of) the asymptotic stationary process dynamics from initial nonstationary data sequences (Section 4). 3. A method, called "binless learning" or "binless OOMs" to estimate from continuous-valued initial nonstationary training sequences a novel kind of descriptive model of asymptotic characteristics of the process' distribution (Section 5). 4. Brief demos of the methods from Section 5 on synthetic data and data derived from simulations of an analytical model of a protein state switching dynamics. In these demos the objective is to estimate asymptotic process properties from training data that start from (very) out-of-equilibrium conditions. These demos reveal a much better compensation for the initial nonstationarity of training data than more traditional HMM models.

Qualitative Assessment

Novelty and relevance The analytical results presented in Section 3 are not new (see my more detailed comments below), though they probably have not yet been presented in exactly this form. I believe that the authors, who apparently come from a modeling community in the biosciences, simply are not aware that it is commonplace wisdom in the OOM field that OOM models of nonstationary processes can be estimated in the same way as stationary ones (as long as the nonstationarity can be described as the effect of a nonequilibrium starting state; stronger forms of nonstationarity where the very observable operators vary with time are not considered in this paper, and have only superficially been investigated by OOM researchers elsewhere). The core contribution of Section 4, an algorithm to recover the stationary state from an OOM estimated from nonstationary data, is novel (but I think it might be unnecessarily complicated, see my detailed comment below). The "binless learning" of "binless OOMs" in Section 5 is entirely novel and quite ingeneous in my view - for me the absolute bummer part in this paper. Judging from the first of the three demos in Section 6, truly amazing inferences can be drawn about asymptotic properties of a stochastic process from highly nonstationary initial-condition-plagued training data (while the remaining two demos I couldn't understand because they require previous knowledge about techniques and terminology from statistical chemistry (?) that is not explained and that I can't summon). Although I am not familiar with the kind of bio/chemistry microphysics applications that the authors use for display, it would think that these binless OOMs constitute an truly impressive and important step of progress. Technical correctness Apart from some minor lapses, mostly notational typos (pointed out below) - fine. Presentation Very good English, crisp command of mathematical formalism. I find that the Introduction over-emphasizes the challenges arising from nonequilibrium data -- these are really solved and easy issues as long as the nonstationarity can be captured by non-equilibrium starting state (the case considered in this article). In contrast, the Intro under-sells in my view the novel modeling opportunities afforded by binless OOM learning. Summary assessment and recommendation A very interesting and innovative contribution -- namely, in Section 5, concerning "binless OOMs". The novelty of Sections 3-4 is marginal, the authors are probably not aware of the existing common understanding of the matters discussed in these Sections (the novel algorithm in Appendix A.4 excepted). If this were a journal paper submission I would recommend a substantial revision with a much deflated treatment of the Section 3-4 material and a much more prominent role of the binless idea. The last two demos are useless because they are unintelligible due to compression - in a revision they would need to be expanded. ********** more detailed comments ********** Lines 28 ff: the article presents the estimation of OOMs from nonstationary data as something that has been rarely done before ("a major challenge … In most literature the observation data are assumed to be equilibrium…"). That is not true. Even in the earliest OOM publications (H. Jaeger(1998): Observable operator models of stochastic processes: a tutorial. GMD Report 42, German National Research Center for Information Technology 1998) it was explicitly recognized that the standard OOM estimation procedure works for stationary and nonstationary data alike. I dare say that this is and always has been obvious to researchers active in this field. In the recent review paper of Thon (Thon, M., Jaeger, H. (2015): Links Between Multiplicity Automata, Observable Operator Models and Predictive State Representations -- a Unified Learning Framework. Journal of Machine Learning Research 16, 103-147) the assumption of stationarity is not even introduced. In other words, Theorem 1 cannot be considered new. Only the generalization 1 (line 129) adds a new aspect to the known picture. Section 4: the initial text paragraph confuses me. It starts by saying that "the only remaining problem for learning OOMs from nonequilibrium data is how to estimate initial state vectors". But then the text proceeds to state that this problem is uninteresting and instead one should try to find the equilibrium state. I am further confused by the fact that in line 149 it is not explained what \hat{omega} is intended to be - is it intended to be the equilibrium state? then it should be written \hat{\bar{\omega}}. -- On a side note, it is confusing that the index _{eq} is used to denote an 'equivalent' OOM, and the \bar to denote an equilibrium. Both words start with 'eq' and in fact I lost one hour getting myself un-trapped from this duplicity. Please consider changing the notation. Lines 148 ff: I don't understand why the equilibrium state \hat{\omega} (or maybe in better notation \hat{\bar{\omega}}) isn't simply calculated as an eigenvector of \hat{Xi}(\mathcal{O}) of largest absolute eigenvalue. Furthermore, some thought should be given to processes that have more than one ergodic component. Then there is no uniquely defined equilibrium state. Section 5: the choice of terminology "binless learning" or "binless OOM" should be explained/motivated. I never heard of this concept before and it is fun to see what Google queries try to retrieve (zero hits on these key phrase, alternative suggestions: "boneless ham", "boneless BBQ"…). The cover story for motivation the material from Section 5 should be phrased more carefully. Text in the Introduction suggests an alternative to parametric / Hilbert space embedding OOMs, but Section 5 only offers a model of asymptotic characteristics of a process, not a model of the process, so the juxtaposition to those continuous-valued OOM techniques is misleading. Introduction should be rephrased carefully. It took me a little effort to understand the mathematical type of the object defined in Eqn (19). It would help the reader to explicitly state that this is a (point) probability measure on \mathcal{O} \times \mathbb{R}^D_1 \times \mathbb{R}^D_2. Line 173f: "where feature functions can be selected as splines, radial basis functions or other …": this needs more explanation. How can e.g. a spline function, applied to L points, give a real number as a feature value? Simulation studies reported in Figs 3 and 4: impossible to understand in this super-condensed presentation. Readers are required to know what TIC analysis is and what w_{TICA} is. The NIPS readership will be lost. Please invest an explanatory paragraph. ********* minor comments / edits ********* line 90, first formal expression, counter index should range over n not s Figure 1: indices in circles should read t-L etc, not t-s Assumption 2 page 4: please specify the range of the f_t functions. I assume it's \mathbb{R}? (or re-use notation \phi from section 3.1) It would be helpful for the reader to state at some early point that nonstationarity of a rank-m process represented by an m-dimensional OOM is equivalent to \omega \neq \omega_{eq} \Xi(\mathcal{O}). line 149 typo "quadrantic" line 192: principle -> principal line 198: "for general g with" incomplete or "with" superfluous

Confidence in this Review

3-Expert (read the paper in detail, know the area, quite certain of my opinion)